# The role of religious narratives and religious orientation towards concerns for the natural environment and animal welfare

Dexon Pasaribu [1☯*], Bagus Takwin [2☯], Pim Martens[3☯]

1 Maastricht Sustainable Institute, Maastricht University, Maastricht, The Netherlands, 2 Faculty of Psychology, Universitas Indonesia, Depok, Indonesia, 3 University College Venlo, Maastricht University, Maastricht, The Netherlands

☯ These authors contributed equally to this work.
* dexon.pasaribu@maastrichtuniversity.nl, dexon.pasaribu@gmail.com

**Data Availability Statement:** All relevant data are within the paper and its Supporting Information files.

**Funding:** The work has been made possible by Government of Indonesia Ministry of Finance's

## Abstract

Several studies show that religion hinders concerns for the natural environment preservation. Others, however, have found that the belief in God or the identification with a particular religion is not associated with measures for environmental concerns. This study investigates the influence of religious narrative framing and the relation between Allport's intrinsic personal (IP) and extrinsic social (ES) religious orientation towards general environmental apathy (GEA) and acceptability for harming animals (AIS). This study surveyed 657 teachers and school staff in East Java, Indonesia. Using ANOVA, we find that religious narrative affects participant's GEA and AIS. Participants in stewardship narrative group have significantly lower GEA and AIS compared to participants in human dominance and the non-narratives control group. Using multiple regression, we also confirm the persistence of religious narrative's influence towards GEA. In addition, lower GEA and AIS correlate with higher IP and lower ES. Lastly, we identify and discuss significant demographic and other determinants relation to GEA and AIS.

## 1 Introduction

Sustainable Development first became prominent in the 1980s with its most mainstream definition of "development that meets the needs of the present without compromising the ability of future generations to meet their own needs" [1]. From this definition, sustainability studies formulate three-pillars consist of social, economic, and environmental sustainability as the objective for sustainable development. From there, the environmental sustainability becomes the domain of sustainability sciences while the former two (namely, economic, and social sustainability) become the domain of development studies. However, a complete reconciliation between both studies is not without challenge. According to Goodland & Daly [2], one of the challenge lies in the difference in both disciplines to prioritize differing aspects of development. While the development goals are fundamentally important, they are quite different from the goals of environmental sustainability, which is the unimpaired maintenance of human life-

Endowment Fund for Education Institution's (LPDP) (Grant no 2017022221010339) scholarship and funding (https://www.lpdp.kemenkeu.go.id). The funder had no role in study design, data collection and analysis, decision to publish, or preparation of the manuscript.

**Competing interests:** The authors have declared that no competing interests exist.

support systems [3] (p. 5). Goodland & Daly [2] differentiate, at the very least, four kinds of capital, which are human-made capital (financial and economic accounts); natural capital (the stock of environmentally provided assets such as soil, atmosphere, forests, water, wetlands); human capital (investments in education, health and nutrition of individuals); and social capital (the institutional and cultural basis for a society to function).

Goodland & Daly [2] challenge the notion of throughput growth in the context of finite earth, in which as a subsystem of the finite and non-growing earth, the economy must eventually adapt to the finite earth. Thus, environmental sustainability requires maintaining the natural capital and to understand sustainability must include both the definition of "natural capital" and "maintenance of resources" (or at least "non-declining levels of resources"). Sustainability means maintaining environmental assets, or at least not depleting them. Goodland & Daly [2] argue that the limiting factor for much economic development has become natural capital as much as human-made capital. "In some cases, like marine fishing, it has become the limiting factor—fish have become limiting, rather than fishing boats. Timber is limited by remaining forests, not by sawmills; petroleum is limited by geological deposits and atmospheric capacity to absorb $CO_2$, not by refining capacity" [2] (p. 1005). Goodland & Daly [2] conclude that eventually, natural capital will limit this cultivated natural capital.

In support to Goodland [3] and Goodland & Daly [2], the present study brings forth the dilemma between sustainability science and development studies where they haven't yet reach consensus on the attainable priorities path-ways of whether to reach environmental sustainability or more anthropocentric (social and economic) sustainability. In one polar there is the urgent need to preserve the natural environment for the sake of itself to recuperate (ecocentric). However, on the opposite polar, there are huge challenges of social and economical needs for sustaining human lives (anthropocentric).

Of this ecocentric-anthropocentric polars, similarly, Thompson & Barton [4] formulate and develop two underlying motives of environmental attitudes, which are ecocentrism—valuing nature for its own sake; and anthropocentrism—valuing nature because of the material or physical benefits it provides; and also, with an additional dimension of general apathy towards the environment [5]. Thompson & Barton [4] propose that the motives and values which underlie environmental attitudes are of great significance in which the same positive attitude in valuing the importance of conserving the natural environment may come from ecocentric or anthropocentric motives, or even both. This is especially relevant after Bjerke and Kaltenborn [5] further riddled this topic when they found that ecocentric motive is scored differently to different job-groups categorization when valuing carnivores animals compared to herbivores. In their study of ecocentric and anthropocentric motives relationship to attitude towards large carnivores, Bjerke and Kaltenborn [5] highlighted that high ecocentrism and low apathy to the natural environment only specifically resonate to those research biologist and wildlife managers groups who score more positive attitude towards carnivores.

For sustainability and the attitude or concerns to the natural environment, White's [6] thesis stating that religion hinders concerns for the natural environment highlights the urge for sustainability efforts as a response to industrial and economic development and growth at that time. White [6] argues that, to some extent, the current ecological crisis is due to the disconnection of nature and spirituality often promoted by religion which gives the human species rights and dominance to exploit nature which in return, forms the basis for exploiting the natural world. Before White [6], religion wasn't considered as a factor contributing to environmental degradation [7]. Many studies after White [6], show interrelation between religion and ecology. Several studies show that less concern for global warming and the environment relates to religious affiliation [8], religious literalism and aspects of religiosity expression [9] and end of the world belief (end-times theology) [10]. On the other hand, there are also studies which

conclude no association between environmental concerns and religious belief and identification [11–13]. One particular study by Smith and Leiserowitz [14] further complicates the relationship between religion and the environment. They find that compared to non-evangelist, the evangelicals are less likely to believe global warming is happening, caused by human activities, and are less worried about it. However, they find that egoistic, altruistic, and biospheric concerns of global warming as the single strongest predictor of evangelical risk assessments and policy support rather than the religious aspects. They add that evangelicals who hold an individualistic worldview are more likely to oppose policies designed to mitigate global warming, and conclude that affect based value orientations, ideologies and worldviews are more important for understanding these divided positions towards global warming than theology per se [14].

Similarly, in animal welfare studies, the role of religion remains unclear. Negative attitudes and behavior toward animals is not significantly related to religion or religious practice [15,16], political ideology [17] and religious ideology [18]. However, increased evidence for the role of religion in the field of animal welfare emerge. Animal treatment studies show that negative attitude towards animals relates to gender, church attendance [19,20] and Christianity as a source of inspiration [21]. In other study, more negative attitude towards animals and more acceptability for harming animals are more likely adopted by practitioner of any religion [22] and relate to the value and relevance of specific animals in different religions [23]; to the types, kinds and species of the said animals [24]; and to religious affiliation and liberal-conservative theological aspect of the affiliated church [25].

Inconsistent role of religion may originated from differing aspects of religion utilized in above studies. Some studies address communication framing of religious messages [14,26], while others address the religious scriptures contents and interpretation [27–29]. Regarding the latter, religious followers' interpretation toward their religion, religious scripture and teachings are somewhat unpredictable considering that it is a factor that widely varies depending on a subject's interpretation and context for re-interpretation [26,30]. Using Feinberg & Willer [30] research as an example, reframing environmental discourse into a specific religious interpretation reduces the gap of environmental concerns between liberals and conservatives. Feinberg & Willer [30] argues that presenting conservatives with pro-environmental messages couched within a set of particularly conservative moral domain, leads conservatives to adopt more pro-environmental attitudes, comparable to those of liberals. In the study of Christian's public voices in the US public debates, Wardekker et al. [26] finds three different narratives addressing fundamental ethical questions, which are 'conservational stewardship' (conserving the 'garden of God' as it was created), 'developmental stewardship' (turning the wilderness into a garden as it should become), and 'developmental preservation' (God's creation is good and changing; progress and preservation should be combined). Wardekker et al. suggest that religious framing of climate change to resonate with the electorates of both progressive and conservative politicians can serve as bridging devices for bipartisan climate-policy initiatives [26].

In addition, from another distant topic, evidence from Indonesia family planning programme shows that between the 1960s and the 1990s, religion played important part in assisting government's family planning programme and policy in Indonesia. By reframing religious scripture, religious influential figures at that time helped to accommodate a more acceptable interpretation for family planning [31]. Religious teachings were reframed to mediate public acceptance for family planning [32]. Indonesia's family planning case study suggests that one of the keys to a change in thinking (and public acceptance in general) is Islamic institutions' joint efforts in reformulating the religious teachings of the shari'a combined with secular sources. These sources ranging from domestic and foreign medical authorities, government

ministries, up to the family planning board of coordination, along with experts from the World Health Organization and other international organizations [33]. Furthermore, although religion alone is not always a good indicator of the religious-political culture, there are wide variations of Islamic shari'a interpretation for contraceptive at that time [34]. These various interpretations then become both the subjects and entry points for an alternative framing of religious scriptures, on one hand defusing some opposing religious views (e.g. the view that limiting the amount of child in one's own family is opposite to God's will and blessings) while on the other hand, promoting religious views that support family planning [35,36]. Thus, framing religious teachings and scriptures into various narratives is one of the main variables of this study.

Observing different definition for the interchangeably terminology of religiosity, religious-ness or religious belief may help to explain various opposing results from abundance of religious studies toward the ecology. Different aspects such as (1) human cognitive, (2) affect, and (3) behavior, such as church attendance, or affiliation [37] in formulating definition for religiosity or religiousness may suggest different angle of defining the variable and consequently, producing different results. Gallagher & Tierney [38] argue that religiosity and religiousness are interchangeable as far an individual's conviction, devotion and veneration towards a divinity is concerned. As the continuation of Pasaribu, Takwin and Martens' research [39], this study also chooses Allport's religious orientation of intrinsic and extrinsic motivation in practicing religious belief. Religious orientation consists of three components. *First*, intrinsic personal (IP) orientation, where religion is personal and people intrinsically motivated and committed to their religious belief and practices in their daily life in trying to follow the moral code of their religion. *Second*, extrinsic personal (EP) religious orientation, where people feel extrinsically motivated by acquiring emotional sense of peace and comfort as the result of doing religious activity. *Lastly*, extrinsic social (ES) orientation in which people perceive social advantages by having a sense of membership from a religious group such as, protection, consolation or social status[40–44]. In examining the relation between animal protection issue with ethical ideologies and religious orientation, Pasaribu, Martens & Takwin [39] find that participants with higher score of IP are more likely to have lower acceptability for harming animals. However, they also find that participants with higher score of ES tend to also score higher acceptability for harming animals.

This study is the continuation of Pasaribu, Martens and Takwin [39], to further examine religious orientation roles toward acceptability for harming animals and concerns for the natural environment by adding religious narratives as the representation of various dogmatic teachings [26,30]. Pasaribu, Martens, and Takwin [39] argue that extrinsic social religious orientation closely relates to social identity from Tajfel's theory of social identity in ways that religious group offers a sense of group positioning within which individuals identify themselves vis-à-vis religious outgroups [45,46]. For religious orientation, the present study focuses the attention to ES, to investigate whether it has diminishing or increasing role when considering one other variable that is religious narrative.

On the other hand, focusing on various religious narratives as a treatment variable in a quasi-experimental design, the present study tries to measure attitude towards the natural environment, religious orientation, and animal importance and to examine any differences between narratives groups. This study investigates the influence of religious teachings to the attitude towards the importance of the natural environment and animal by reframing religious scriptures into two priming narratives which are stewardship (SN) and human dominance narrative (DN) as group treatments. In SN, we compile religious scriptures which stress the importance for human to safeguard the nature. In DN, we collect religious texts which stress the notion of human being as the most noble amongst all creation of God. Thus, we expect

that respondents assigned to read SN will be more likely to score a lower apathy toward the natural environment and a lower acceptability for harming animals compared to those assigned to DN, leaving the control group (the group without any narrative treatment) somewhere in between. Afterwards, we also investigate general demographic determinants such as gender, age, education, household income, meat consumption, pet ownership, and religious organization affiliation [21,47].

As the continuation of Pasaribu, Martens and Takwin [39], this study uses the same sample, purposefully selected from the population in East Java province. Other than considering that it represents the oldest, most influential Islamic organizations (e.g., Nahdlatul Ulama and Muhammadiyah), the province of East Java is also one of the provinces with the most diverse Islamic denomination. Also, religion matters in Indonesia's social foundation. For one instance, looking at Nahdlatul Ulama (NU), not only that it is the largest Islamic organization in Indonesia, it also has political influence from regency-level al the way up to the national level [48]. From Nahdlatul Ulama, Indonesia had Abdurrahman Wahid as the fourth President of Indonesia (1999–2001) and currently, Ma'ruf Amin as the current Indonesian vice president (took office in 2019). Thus, the present study views that Indonesia, especially East Java, as the perfect population where religion matters and strongly influences social and political constellation of Indonesia. However, by taking samples from East Java province, the present study avoids over-generalization of the results.

## 2 Materials and methods

We confirm that Maastricht University's institutional review board (ethics committee) reviews and approve this article. We have submitted the plan for conducting the study, the time schedule, the questionnaires, and the tools for collecting data and acquired the approval from the Maastricht University's Ethics Review Committee Inner City faculties. This research article conforms ethics for human participant regulated by the General Rules for Information Protection (European Union) 2016/679. We carefully manage all personal information so that personal data are safe from third parties and stored on servers that are not accessible to public. We replace names and position with an alphanumeric code to keep participants' identity protected.

The present study targets school teacher and staff in considering that schools are subjects to nation-wide education curriculum [49]. School teachers also have important roles of transformative intellectuals [50–52], and authoritative figures which may provide insight of moral, political and ethical issues to their students [53,54]. We made written invitation to each school requesting for participation. Relevant body of Indonesia government ranging from national, province to districts. Legalized by the invitation letter. There were schools that decline to participate while schools or universities that accept the invitation were surveyed. Before surveying, we request oral consent from each of the participant to re-confirm that they are voluntarily participating in the survey.

We invite sixty-seven schools (ranging from junior to senior high schools) to participate in the survey. The invitation request all participating schools to provide a balanced proportion of male and female teachers or school staff. Total of 37 schools, from ten districts of East Java, replied and agreed to participate, providing 1007 participants. However, only 929 participants were analyzed due to the removal of seventy-eight participants because of incomplete and unengaged answers. Normal distribution, factor analysis and Cronbach alpha's reliability were analyzed using this 929 participants data. However, after cross-checking with priming narratives treatment check questionnaires, we further remove 272 participants who score below the expected mean (*see* section 3.2) from both stewardship and human dominance narrative groups, leaving a total of 657 participants for the ANOVA and multiple regression analysis.

The main variable of this research are religious priming narratives (as the representation of religious scripture and teachings), religious orientation, attitude toward the importance and conservation of the natural environment, and the acceptability for harming animals. The present study uses quasi-experimental comparative design to analyze various treatments of religious narrative priming toward religious orientation, attitude towards the importance of the natural environment and attitude towards the importance of animal protection. With the help from religious leaders in East Java, Indonesia, we develop two distinct narratives. The first narrative synthesizes religious teachings which support environment preservation (stewardship narrative) while the second synthesizes religious teachings which inform the privilege of human beings as the ultimate creation (human dominance narrative). Using these narratives, we implement three different treatment groups. We give one group the stewardship narrative (SN), and the second group the human dominance narratives (DN). The third group is a control group, where respondents directly fill the survey without reading any priming narrative.

All questionnaires in the survey is originally in English (S2 File, questionnaires in English) and we translate them to Bahasa Indonesia (S3 File, questionnaires Bahasa Indonesia adaptation). We use expert judgement for the method of translation, and back translation. We translate the questionnaires to Bahasa Indonesia and sent to experts for evaluation and finalization of the translation. After corrections, three Indonesian academicians from Universitas Indonesia back translate all the questionnaires to English. We retain back-translated items that are similar to the English version, and modify or delete those items which lack similarity. The set of questionnaires consist of six sections. In the first section, we asked a variety of important determinants and demographic details such as birth year (age), gender, highest level of education completed, their experience or participation in either animal protection, nature conservation, or human health organization, their household composition (for example, single, married, or widow(er), with children or not), place of residence (rural or urban), type of house (apartment, live with parents, etc.), their opinion regarding the importance of religion/spirituality in their lives, household income, pet ownership, kinds of pet, their weekly frequency of meat consumption, and the frequency of visiting public zoos or aquariums in a year.

In the second section, we give participants three types of treatment narratives. We develop two types of religious narratives based on religious scriptures. The first is environmental stewardship narrative, in which we collect religious scriptures which stressed the importance for human to safeguard the nature. The second is the human dominance narratives in which we collect religious texts which stressed the notion of human being as the most noble amongst all creation of God. From these two narratives (S1 File, Treatment narratives), we divide participants into three group treatments namely stewardship narrative (SN), dominance narrative (DN), and control group where participants fill the survey without given any narratives (NN). Except NN group, we oblige participants to read the narrative presented to them before filling the questionnaires.

The third section is a questionnaire we developed as a group treatment check. Not necessarily opposed to each other, each question is a pair of statement of which respondents need to choose one they prefer the most. These questions measure whether the participants read the narratives before filling the survey. Using this, we first examine whether the priming narrative treatment applies to each of the respondent in their treatment group. We develop ten questions as a treatment check to filter unengaged respondents (those participants who may decide to fill the questionnaires without reading the narrative) from those who read the priming narratives before taking the survey. Each question refers to what written in the narratives using 4-point semantic differential type of response. A mean score above 2.5 suggest that the respondents read the narratives properly and therefore included in analysis. A mean score equal to or

below 2.5 suggest that respondents did not read the priming narratives and therefore removed from the ANOVA and multiple regression analysis. Table 1 list all the questions in this section.

In the fourth section, Thompson & Barton's [4] Ecocentric-Anthropocentric Scale of Environmental Attitude (EASEA) is used to measure environmental attitudes which consists of three dimensions, namely ecocentrism, anthropocentrism and general environment apathy. Amérigo et al. [55] argue that ecocentrism seems to include two concepts: the self in nature (egobiosphere) and nature itself (biosphere). In ecocentrism motives, on the one hand, there are items about physical or psychological benefits for the individual, brought about by the mere fact of being in or thinking about nature (e.g., "Being out in nature is a great stress reducer for me"). These are related to the positive emotional effects produced by contact with nature where the protagonist is the self and it is the only direct beneficiary of the goodness of the natural environment which could be considered to be related to an egoistic dimension [55]. On the other hand, the remaining ecocentric items refer to biospheric aspects that emphasize the intrinsic value of Nature (e.g. "Nature is valuable for its own sake") which may be oriented into two different viewpoints of (a) a psychosocial perspective that contemplates the human-being-in-nature and in which the environment is valued as an element that procures the individual's physical and psychological well- being, and (b) a strictly biospheric dimension in which the environment is valued intrinsically and that contemplates the nonhuman elements of nature [55]. In short, Amérigo et al. [55] propose that there are anthropocentric valuation in the Thompson & Barton's [4] ecocentric scale. Thus, it is exceedingly difficult to differentiate whether environmental concerns stem from ecocentric or anthropocentric motives or both. Therefore, this study only uses the general environmental apathy scale to measure whether a person has apathy disposition towards their natural environment. From total 31 items of Thompson & Barton's [4] Ecocentric-Anthropocentric Scale of Environmental Attitude (EASEA), there are 12 questions measuring ecocentric motive 10 questions measuring anthropocentric motive of nine questions measuring general apathy toward the environment. As explained above, this study only uses the general environment apathy (GEA) scale. Examples of the statements are, 'environmental threats such as deforestation and ozone

**Table 1. Priming narrative treatment check questions.**

| No | Statement | | |
|---|---|---|---|
| 01. | Human beings are superior to other beings | ○ ○ ○ ○ | Environmental damage is high and worrying |
| 02. | Humans are sent to lead | ○ ○ ○ ○ | natural resources are threatened |
| 03. | Human beings are superior to other beings | ○ ○ ○ ○ | God sent man to take care of |
| 04. | Humans needs to support their family | ○ ○ ○ ○ | Humans need to protect and care for nature and the environment |
| 05. | God sent man to lead | ○ ○ ○ ○ | Wildlife welfare (protecting all living things) |
| 06. | God sent man to take care of | ○ ○ ○ ○ | Humans are sent to lead |
| 07. | God sent man to lead | ○ ○ ○ ○ | Animals need a place to live |
| 08. | Humans don't give in and worry too much | ○ ○ ○ ○ | Wildlife welfare (protecting all living things) |
| 09. | God created humans as noble creatures | ○ ○ ○ ○ | God sent man to take care of |
| 10. | God sent man to lead | ○ ○ ○ ○ | Human life depends on nature |

depletion have been exaggerated, 'too much emphasis has been placed on conservation,' or 'I don't care about environmental problems.' However, after principal axis factoring factor analysis (S4 File, Missing case analysis, Factor Analysis and Reliability; Tables 10–18), this study reduced the items to only seven items, and found that GEA consists of two factors instead of one. However, after confirming a good Cronbach alpha's reliability in one factor model, the present study decided to retain the environmental apathy dimension as it was originally, a one factor construct (model Two, *see* Table 2).

In the fifth section, the Animal Issue Scale (AIS) [16] is used to measure acceptability toward harming animals. There are forty-three questions in the original AIS, representing eight animal issues: use of animals, animal integrity destruction, killing animals, animal welfare deprivation, experimentation on animals, changes in animals' genotypes, harm animals for environmental reasons, and societal attitudes toward animals [harm animals for social issues]. Respondents rate each question on a five-point scale ranging from one, extremely unacceptable, to five, extremely acceptable. A high score on a question indicates a high level of acceptability for the particular issue [56]. Using principal axis factoring factor analysis (S4 File, Missing case analysis, Factor Analysis and Reliability; Tables 3–5), the original 'killing animal' and 'animal deprivation' issues were identified as one factor (Table 3). Thus, the present study reduced AIS to only thirty-one items, conveyed only 7 factors. AIS score is the mean score from all thirty-one items.

In the sixth and final section, the Religious Orientation Scale (ROS) [41,57,58] was originally used to measure intrinsic and extrinsic religious orientation. We used Maltby's [44] 15-item version which incorporated Kirkpatrick's [59] analysis expanding ROS into three scales: intrinsic orientation (IP), extrinsic personal—religion as a source of comfort (EP) and extrinsic social—religion as social gain (ES). The 15-item scale therefore consists of nine questions addressing IP, for example, 'I try hard to live all my life according to my religious beliefs', 'My whole approach to life is based on my religion', 'It is important to me to spend time in private thought and prayer'); three questions addressing EP, for example 'Prayer is for peace and happiness', 'I pray mainly to gain relief and protection'; and lastly, the remaining three covering the ES dimension, for example, 'I go to church because it helps me make friends', 'I go to church mainly because I enjoy seeing people I know there'. However, after principal axis

**Table 2. EASEA-general environment apathy rotated factor matrix.**

| Items | Model 1 (using eigen value > 1) | | Model 2 (as one factor) |
|---|---|---|---|
| | **1** | **2** | |
| ECCANTH03 Environmental threats such as deforestation and ozone depletion have been exaggerated | .462 | | .518 |
| ECCANTH07, it seems to me that most conservationists are pessimistic and somewhat paranoid. | .535 | | .594 |
| ECCANTH09, I do not think the problem of depletion of natural resources is as bad as many people make it out to be | .692 | | .651 |
| ECCANTH10, I find it hard to get too concerned about environmental issues | .721 | | .611 |
| ECCANTH14, I do not feel that humans are dependent on nature to survive | | .445 | .545 |
| ECCANTH17, I don't care about environmental problems | | .746 | .549 |
| ECCANTH18 I'm opposed to programs to preserve wilderness, reduce pollution, and conserve resources | | .683 | .591 |

Extraction Method: Principal Axis Factoring. Rotation Method: Varimax with Kaiser Normalization.

**Table 3. AIS rotated factor matrix.**

| | Factor | | | | | | |
|---|---|---|---|---|---|---|---|
| | 1 | 2 | 3 | 4 | 5 | 6 | 7 |
| AI01_AnimUse Keeping animals for the production of food or clothing | | | | | | | .490 |
| AI02_AnimUse Keeping animals as pets | | | | | | | .447 |
| AI04_AnimUse Using animals for work | | | | | | | .624 |
| AI05_AnimUse Using animals for entertainment or sports | | | | | | | .654 |
| AI08_Intgrty De-sexing by hormone implants | | | | .542 | | | |
| AI09_Intgrty Removal of a body part, such as tail docking or de-clawing | | | | .662 | | | |
| AI10_Intgrty Marking animals by branding or ear notching | | | | .589 | | | |
| AI11_Intgrty Removal of dead tissue, such as hair/wool removal or foot trimming | | | | .557 | | | |
| AI14_Kill Using animals for products after their natural death | .439 | | | | | | |
| AI16_Kill Euthanizing healthy and unwanted pets because of overpopulation | .556 | | | | | | |
| AI17_Welfare Depriving animals of their needs for food and water | .768 | | | | | | |
| AI18_Welfare Depriving animals of an appropriate environment to rest, including shelter | .765 | | | | | | |
| AI19_Welfare Inflicting pain, injury, or disease on animals | .798 | | | | | | |
| AI20_Welfare Not providing sufficient space, proper facilities and company needed for animals | .701 | | | | | | |
| AI21_Welfare Subjecting animals to conditions and treatment which cause mental suffering | .501 | | | | | | |
| AI24_Xprmnt Medical experiments using animals to improve human health | | | | | .553 | | |
| AI25_Xprmnt Testing cosmetics or household products on animals | | | | | .636 | | |
| AI26_Xprmnt Operating on living animals for the benefits of human medicine research | | | | | .755 | | |
| AI27_Genchng Increasing animals' reproductive or productive capabilities by genetic changes, e.g. cows producing more milk | | | .633 | | | | |
| AI28_Genchng Increasing animals' health or disease resistance by genetic changes | | | .693 | | | | |
| AI29_Genchng Creating farm animals that are more profitable because they feel happy with little stimulation and have little desire to be active | | | .749 | | | | |
| AI30_Genchng Genetic selection of pet animals, such as dogs and cats, to increase their rarity, potential for showing or pedigree value | | | .600 | | | | |
| AI34_EnvIss Controlling wildlife populations by killing | | | | | | .542 | |
| AI35_EnvIss Controlling animal populations by sterilization | | | | | | .439 | |
| AI36_EnvIss Destroying the habitat of endangered animal species | | | | | | .596 | |
| AI37_EnvIss Destroying the habitat of non-endangered animal species to develop and promote urbanization or crops to feed humans | | | | | | .465 | |
| AI39_SocAtt Considering some animal species as sacred or good luck symbols or totems | | .606 | | | | | |
| AI40_SocAtt Considering some animal species as evil or bad luck | | .765 | | | | | |
| AI41_SocAtt Parents displaying cruel treatment of animals in front of their children | | .591 | | | | | |
| AI42_SocAtt Inflicting pain or injury on animals as part of cultural traditions | | .570 | | | | | |
| AI43_SocAtt Cloning animals for human benefit | | .435 | | | | | |

Extraction Method: Principal Axis Factoring. Rotation Method: Varimax with Kaiser Normalization.

factoring factor analysis (S4 File, Missing case analysis, Factor Analysis and Reliability; Table 21–26), the present study found only two dimensions of intrinsic personal (IP) and extrinsic social (ES). After factor analysis, this study considered EP and IP as one factor (Table 4).

## Statistical analysis

Scores of religious orientation and motives toward environmental protection and acceptability for harming animals were analyzed with IBM SPSS 24 Statistical software. We use Analysis of variance (ANOVA, with Bonferroni correction) to measure the various treatments that may

**Table 4. ROS rotated factor matrix.**

| | Factor | |
|---|---|---|
| | **1** | **2** |
| ROS01 (IP) I try hard to live all my life according to my religious beliefs | .673 | |
| ROS03 (IP) I have often had a strong sense of God's presence | .608 | |
| ROS04 (IP) My whole approach to life is based on my religion | .705 | |
| ROS05 (IP) Prayers I say when I'm alone are as important as those I say in church | .577 | |
| ROS06 (IP) I attend church once a week or more | .358 | |
| ROS07 (IP) My religion is important because it answers many questions about the meaning of life | .741 | |
| ROS08 (IP) I enjoy reading about my religion | .750 | |
| ROS09 (IP) It is important to me to spend time in private thought and prayer | .630 | |
| ROS10 (EP) What religion offers me most is comfort in times of trouble and sorrow | .665 | |
| ROS11 (EP) Prayer is for peace and happiness | .764 | |
| ROS12 (EP) I pray mainly to gain relief and protection | .622 | |
| ROS13 (ES) I go to church because it helps me make friends | | .833 |
| ROS14 (ES) I go to church mainly because I enjoy seeing people, I know there | | .894 |
| ROS15 (ES) I go to church mostly to spend time with my friends | | .787 |

Extraction Method: Principal Axis Factoring. Rotation Method: Varimax with Kaiser Normalization.

affect respondents' concerns toward the natural environment and animal protection. This study also used Pearson's correlation product moment in investigating the relation between general environment apathy and acceptability for harming animals.

As Pearson correlation procedure is vulnerable from skewed and kurtosis distribution, we made preliminary normal distribution check to avoid inflated correlation. We check each item in the questionnaire for normality (S4 File, Missing case analysis, Factor Analysis and Reliability; Table 2). In regards to normal distribution assumption, Kim [60] stressed that the tendency of large samples producing inflated z in consideration to large samples will usually produce a very small standard error for both skewness and kurtosis. Therefore, using skewness and kurtosis reference values for N more than 300, the present study removed items with kurtosis value outside the range between -7 to 7, or skew value outside the range between -2 to 2

**Table 5. Skewness and kurtosis value of main variables.**

| | N | Skewness | | Kurtosis | |
|---|---|---|---|---|---|
| | **Statistic** | **Statistic** | **Std. Error** | **Statistic** | **Std. Error** |
| General Environmental Apathy (GEA) | 657 | .325 | .095 | -.154 | .190 |
| AIS | 657 | .364 | .095 | .755 | .190 |
| Animal use subscale | 657 | -.036 | .095 | .208 | .190 |
| Integrity destruction | 657 | .470 | .095 | .318 | .190 |
| Killing animal and animal welfare deprivation | 657 | .759 | .095 | .444 | .190 |
| Animal experimentation | 657 | -.190 | .095 | -.029 | .190 |
| Genotype change | 657 | -.435 | .095 | .446 | .190 |
| Harm animal for environmental issue | 657 | .394 | .095 | -.067 | .190 |
| Societal attitude toward animal. | 657 | .547 | .095 | .212 | .190 |
| ROS Intrinsic Personal (IP) | 657 | -.620 | .095 | .427 | .190 |
| ROS_Extrinsic Social (ES) | 657 | .162 | .095 | -.579 | .190 |
| Valid N (listwise) | 657 | | | | |

**Table 6. Descriptive statistics and measurement characteristics for variables.**

| Variable | Scale description | Number of items | Reliability | Mean | SD |
|---|---|---|---|---|---|
| ROS-Intrinsic Personal (IP) | 5-point Likert-like | 11 | 0.88 | 4.23 | 0.53 |
| ROS-Extrinsic social (ES) | 5-point Likert-like | 3 | 0.87 | 2.82 | 1.01 |
| General Environment Apathy (GEA) | 5-point Likert-like | 7 | 0.77 | 2.55 | 0.72 |
| Animal Issue Scale (AIS) | 5-point Likert-like | 31 | 0.92 | 2.57 | 0.53 |
| Animal use | 5-point Likert-like | 4 | 0.66 | 3.13 | 0.66 |
| Integrity destruction | 5-point Likert-like | 4 | 0.78 | 2.43 | 0.80 |
| Killing-welfare deprivation | 5-point Likert-like | 7 | 0.88 | 2.12 | 0.78 |
| Experiment | 5-point Likert-like | 3 | 0.82 | 3.01 | 0.85 |
| Genetic change | 5-point Likert-like | 4 | 0.80 | 3.30 | 0.74 |
| Harm for environmental issues | 5-point Likert-like | 4 | 0.75 | 2.40 | 0.80 |
| Harm for social issues | 5-point Likert-like | 5 | 0.84 | 2.15 | 0.78 |

N = 657.

[60]. Table 5 shows that all scales from the collected data is safely within the normal distribution bound.

## 3 Results

### 3.1 Instrument validity

Table 6 provides the descriptive statistics for the variables used in the analysis. All the Cronbach's coefficient are acceptable, ranging from a good internal consistency value of 0.66 for the 'use of animal' subscale to a value of 0.92 for the overall AIS scale.

The mean score for IP was 4.23 (SD = 0.53, with maximum score of five) indicating that, overall, the respondents mostly expresses strong agreement to items that indicate intrinsic motivation and commitment to their personal religious life. The mean score for ES was 2.82 (SD = 1.01) indicating that, the respondents tend to be undecided to questionnaire statement that indicates religious practices as an instrument for social affiliation. The general environmental apathy mean score was 2.55 (SD = 0.72), indicating that, the respondents mostly express disagreement or neutrality to item that indicate environmental apathy. The mean score of overall acceptability toward harming animal (AIS) was 2.57 (SD = 0.53), indicating that, in general, respondents find that statements about harming animals are unacceptable for them. Except for the issues of animal use (mean of 3.13, SD = 0.66), animal experimentation (mean of 3.01, SD = 0.85) and genetic change (mean of 3.3, SD = 0.74), the remaining subscales of animal integrity destruction (mean of 2.43, SD = 0.80), killing-welfare deprivation of animal (mean of 2.12, SD = 0.78), harm (animals) for environmental issue (mean of 2.40, SD = 0.80), and harm (animals) for social issue (mean of 2.15, SD = 0.78) most of the respondents answered lower acceptability for harming animals.

### 3.2 Response rates

From 1007 total responses obtained, we removed seventy-eight respondents (8%) due to unengaged answers (in other words, these were the respondents who gave the same answer for all the questions in the questionnaire). After the removal, there were still incomplete answers (listwise missing case) in the remaining 929 participants (S4 File, Missing case analysis, Factor Analysis and Reliability, Table 1). We then imputed these incomplete answers using a linear trend method. Afterwards we check for normal distribution, data cleaning, scoring, factor

analysis, and reliability analysis. Lastly, we checked for narratives treatment check total scores and further removed 272 participants (136 participants from SN group and 137 participants from DN group) who scored below expected mean (2.6). We collect and analyze data from total 657 respondents. The final amount of respondent assigned to each treatment groups are 22% to SN (N = 148), 29% to HN (N = 188), and 49% to NN control group (N = 321).

Respondents' mean age is 36 years old (SD = 10), and consists of 51% female (N = 334) and 49% male (N = 323). This study has a balanced amount of participants from rural (60%) and urban (39%) area. For the completed level of education, 72% hold a bachelor's degree, 13% hold an advance Master to PhD degree, while the remaining 15% either graduated diploma, middle or senior high school or did not answer. For home ownership, 56% live in their own a house, 32% still live with their parents, 10% live in a rented room, and 1% live in apartment. Additionally, we gathered information about pet ownership, 52% of respondents didn't have any pet and the remaining 48% of respondents at the very least adopted one pet. We also gathered data about zoo or aquarium visitation, where 43% of the respondents visited public zoo or aquarium once in every two or more years, 23% never visited, 21% once a year, 8% at least once every six months, and lastly 5% visited a zoo once a month, leaving the remaining 1% respondents without an answer. Lastly, 36% of the respondents ate meat two to three days in a week, 29% ate meat once in a week, 13% four to six days in a week, 13% ate meat every day, 7% didn't eat meat, leaving the remaining 1% respondents without answer.

### 3.3 Natural environment preservation attitude (EASE)

In Table 7 we find significant correlation between general environment apathy (GEA) with acceptability for harming animals (AIS) (r[655] = 0.335, $p{<}0.01$).

### 3.4 The role of religious narratives to the attitude towards the importance of natural environment and acceptability toward harming animal

In Table 8, Using ANOVA, a closer inspection to the between-subject effects shows significant between-group differences in environment apathy (F[2] = 5.71, $p$ = 0.003), overall AIS (F[2] = 6.13, $p$ = 0.002), AIS animal integrity destruction issue (F[2] = 5.41, $p$ = 0.005), AIS animal killing and welfare (F[2] = 3.05, $p$ = 0.048), and AIS harming animal for environment issue (F[2] = 4.89, $p$ = 0.008). There are no difference of IP and ES between treatment groups.

Using Bonferroni for post-hoc test, Table 9 showed that there are no significant difference between DN and NN narratives, suggesting that the population represented by the control group has similar apathy for the environment and acceptability for harming animals as DN group. However, compared to the SN group, DN (Mean difference of 0.19, $p$ = 0.049) and NN (Mean difference of 0.24, $p$ = 0.003) have higher environment apathy; and DN (Mean difference of 0.20, $p$ = 0.002) and NN (Mean difference of 0.14, $p$ = 0.024) have higher acceptability for harming animals.

**Table 7. Correlation matrix between EASEA components.**

|  | GEA | AIS |
| --- | --- | --- |
| General Environment Apathy (GEA) |  |  |
| Acceptability for harming animals (AIS) | .335** |  |

N = 657

**Correlation is significant at the 0.01 level (2-tailed)

*Correlation is significant at the 0.05 level (2-tailed).

**Table 8. One-way anova between subject effects tests.**

| Source | Dependent Variable | Type III Sum of Squares | df | Mean Square | F | Sig. |
|---|---|---|---|---|---|---|
| Priming narratives | GEA | 5.883 | 2 | 2.942 | 5.707 | .003 |
| | AIS | 3.417 | 2 | 1.708 | 6.132 | .002 |
| | AIS-Integrity destruction (AIS-ID) | 6.808 | 2 | 3.404 | 5.411 | .005 |
| | AIS-Kill animal and welfare deprivation (AIS-KW) | 3.724 | 2 | 1.862 | 3.048 | .048 |
| | AIS-Harm animal for environment issue (AIS-HEI) | 6.252 | 2 | 3.126 | 4.890 | .008 |
| | Intrinsic personal religious orientation (IP) | 1.218 | 2 | .609 | 2.145 | .118 |
| | Extrinsic social religious orientation (ES) | 2.067 | 2 | 1.034 | 1.017 | .362 |

$N_{SN}$ = 148, $N_{DN}$ = 188, $N_{NN}$ = 321, Total N = 657, For a more detailed results see S5 File, Manova, Tables 1–4.

Specific to each animal issues, compared to the SN group, DN have a higher acceptability for animal integrity destruction issue (Mean difference of 0.27, $p$ = 0.007) and for harming animals for environmental issue (Mean difference of 0.28, $p$ = 0.006) while NN only have a higher acceptability in the animal integrity destruction issue (Mean difference of 0.22, $p$ = 0.015). These results emphasize the influence of stewardship narrative in promoting a lower environmental apathy and acceptability for harming animal to the general population (represented by the control group), which happens to be very similar with the dominance narrative group (Fig 1).

## 3.5 The role of religious orientation to the attitude towards the importance of natural environment and acceptability toward harming animal

In this section, we develop two models and use the multiple regression method for the analysis (Tables 10 and 11). Both model analyze the three main variables relation to AIS, namely intrinsic personal (IP), extrinsic social (ES) religious orientation, and priming narratives. The only

**Table 9. Bonferroni post-hoc test between stewardship and dominance narrative group treatment.**

| Dependent Variable | (I) Priming Narration | (J) Priming Narration | Mean Difference (I-J) | Std. Error | Sig. | 95% CI | |
|---|---|---|---|---|---|---|---|
| | | | | | | Lower Bound | Upper Bound |
| GEA | Stewardship (1) | Human domination (2) | -.1902* | .07890 | .049 | -.3795 | -.0008 |
| | | No Narration (3) | -.2391* | .07133 | .003 | -.4103 | -.0678 |
| | Human domination (2) | No Narration (3) | -.0489 | .06594 | 1.000 | -.2071 | .1094 |
| AIS | Stewardship (1) | Human domination (2) | -.1993* | .05801 | .002 | -.3386 | -.0601 |
| | | No Narration (3) | -.1393* | .05245 | .024 | -.2652 | -.0134 |
| | Human domination (2) | No Narration (3) | .0600 | .04848 | .648 | -.0563 | .1764 |
| AIS-ID | Stewardship (1) | Human domination (2) | -.2681* | .08716 | .007 | -.4773 | -.0589 |
| | | No Narration (3) | -.2222* | .07880 | .015 | -.4114 | -.0331 |
| | Human domination (2) | No Narration (3) | .0458 | .07284 | 1.000 | -.1290 | .2207 |
| AIS-KW | Stewardship (1) | Human domination (2) | -.1967 | .08589 | .067 | -.4028 | .0095 |
| | | No Narration (3) | -.1664 | .07766 | .098 | -.3528 | .0200 |
| | Human domination (2) | No Narration (3) | .0303 | .07178 | 1.000 | -.1420 | .2026 |
| AIS-HEI | Stewardship (1) | Human domination (2) | -.2729* | .08785 | .006 | -.4838 | -.0621 |
| | | No Narration (3) | -.1753 | .07943 | .083 | -.3659 | .0154 |
| | Human domination (2) | No Narration (3) | .0976 | .07342 | .552 | -.0786 | .2739 |

Based on observed means. The error term is Mean Square (Error) = 1.016. For a more detailed results see S5 File, Manova, Table 4.
*. The mean difference is significant at the .05 level.

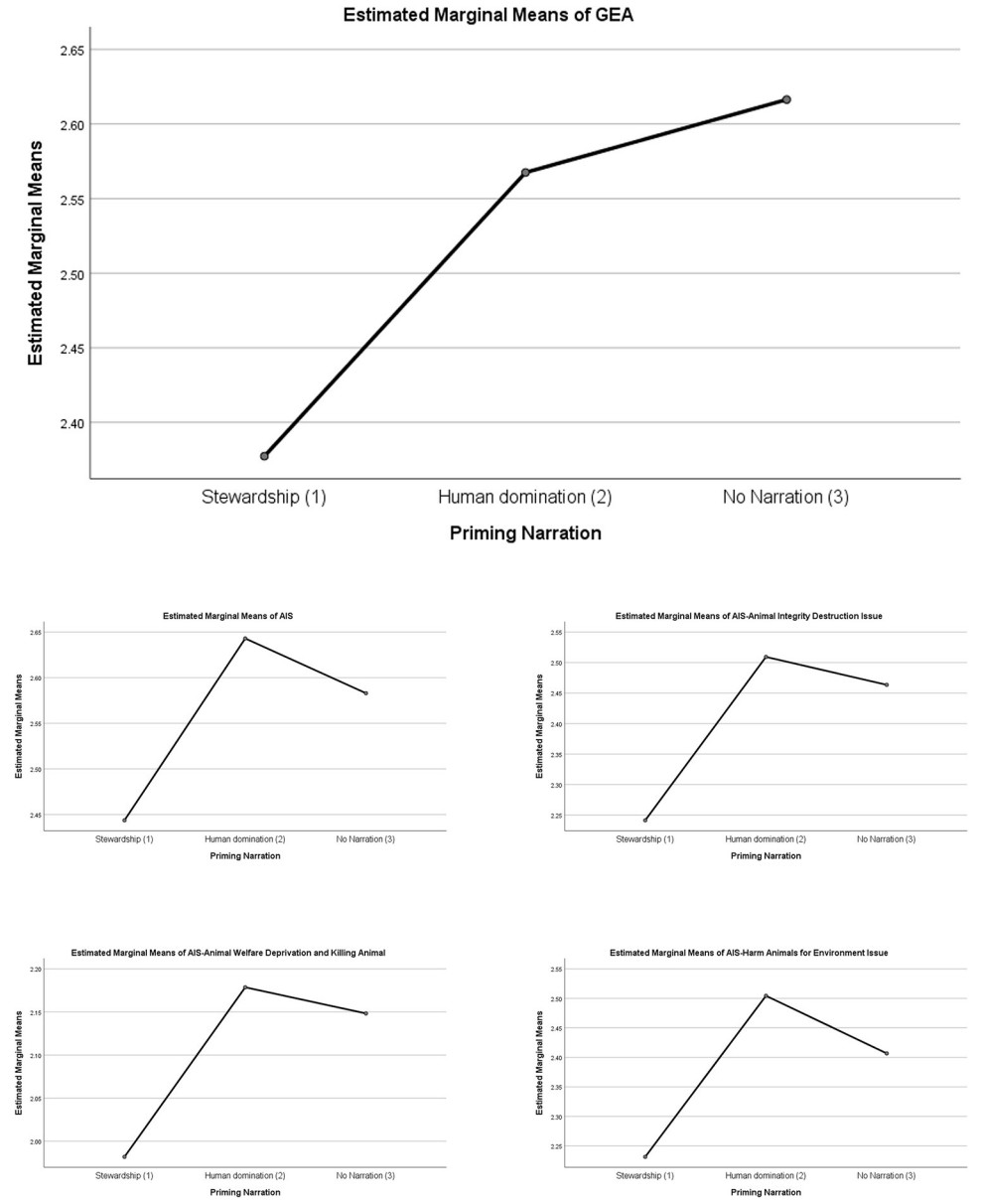

**Fig 1. Differences between various priming narration group treatment.**

difference is that one of the models investigates all the main variables along with demographic and other determinants taken together as well as independently.

Higher IP relates to lower environmental apathy (b = -0.35, p<0.01 in the first model, and b = -0.29, p<0.01 in the second model); and to lower AIS (b = -0.22, p<0.01 in both the first and second model). Moreover, IP relatively has small effect-size for both GEA and AIS. Regarding ES, we find that higher ES relates to higher environmental apathy (b = 0.19, p<0.01 in the first model, and b = 0.14, p<0.01 in the second model); and a higher overall acceptability for harming animals (b = 0.12, p<0.01 in the first, and b = 0.09, p<0.01 in the second model). This means that when holding all other variables constant, one point increase in ES is more likely to increase GEA as much as 0.19 point in the first, and 0.14 in the second model; while

**Table 10. Main variables with demographic and other important determinants to GEA.**

| | | GEA | | | Effect size | | CI (95%) | |
|---|---|---|---|---|---|---|---|---|
| | | b | Std. B | | | | Lower | Upper |
| Model 1: Main variables | | | | | | | | |
| R = 0.386 | (Constant) | 3.55 | | ** | | | 3.12 | 3.98 |
| R2 = 0.149 | Stewardship priming narrative group[1]? Yes (1)–No (0) | -0.24 | -0.14 | ** | 0.33[D] | + | -0.37 | -0.11 |
| df = 4, 652 | ROS Personal | -0.35 | -0.26 | ** | 0.07[C] | + | -0.44 | -0.25 |
| | ROS Social | 0.19 | 0.27 | ** | 0.08[C] | + | 0.14 | 0.24 |
| Model 2: Main variables with demographic and other determinants | | | | | | | | |
| R = 0.465 | (Constant) | 3.14 | | ** | | | 2.33 | 3.95 |
| R2 = 0.216 | ROS Personal | -0.29 | -0.21 | ** | 0.04[C] | + | -0.41 | -0.16 |
| df = 39, 402 | ROS Social | 0.14 | 0.21 | ** | 0.04[C] | + | 0.08 | 0.21 |
| | How often do you consume meat in a week[2]? I don't consume meat: Yes (1)–No (0) | 0.39 | 0.14 | ** | 0.02[C] | | 0.12 | 0.66 |
| | Stewardship priming narrative group[1]? Yes (1)–No (0) | -0.17 | -0.10 | * | 0.33[D] | + | -0.34 | -0.00 |

Unsignificant result omitted. For full results see S6 File, -Multiple regression Tables 1 to 2.

[*]p < .05

[**]p < .01

[A]regression using enter method in a stepwise manner

[B]regression using enter method

[C]effect-size calculation using eta squared (F2); [D]effect-size calculation using Hedge's g; +small effect size F2> = 0.02 (or in some cases of categorical dummy variable, using Cohen's D/Hedges'g > = 0.2)

++medium effect size F2> = 0.15 (or in some cases of categorical dummy variable, using cohen's D/Hedges'g > = 0.5)

[1]compared to respondents who fill the survey without having to read any narrative

[2]compared to respondent who eat meat once a week

also, decrease 0.12 point of AIS score in the first, and 0.09 in the second model. Moreover, ES has small effect-size to both GEA and AIS in both regression models.

Multiple regression results also re-confirm and therefore strengthen the significance of priming narrative treatment groups previously found using ANOVA. Like all categorical variables, for priming narratives treatment variable, we use dummy variables in which each participant scored with either one or zero for both stewardship narrative (SN) and human dominance narrative (DN). In short, in SN dummy variable, we give score one for all participants in the SN group and 0 to DN and NN (control) participants. Likewise, in DN and NN dummy variable., we give score one for all participants in the DN group, and zero to SN and NN (control) participants. It is important to note that there is inherent limitation in using dummy variable method in regression. However, to investigate the relation of all the main variables towards the outcome variables while still taking demographical and other important determinants as well as independently, the dummy variable method is sufficient. Additionally, in measuring the effect-size of any significant categorical dummy variable, we use Hedge g's formula.

In model one, the multiple regression shows that stewardship narrative relates with GEA (b = -0.17, p<0.05) and AIS (b = -0.17, p<0.05). SN group participants are more likely to have a lower GEA and AIS score compared to NN group. However, in the second model, multiple regression only find the relation between SN and GEA (b = -0.17, p<0.05), whereby SN group participants are more likely have a lower GEA score compared to NN group.

**Table 11. Main variables with other important determinants to AIS.**

| | | AIS | | | Effect size | | CI (95%) | |
|---|---|---|---|---|---|---|---|---|
| | | b | Std. B | | | | Lower | Upper |
| Model 1: Main variables | | | | | | | | |
| R = 0.339 | (Constant) | 3.18 | | ** | | | 2.85 | 3.50 |
| R2 = 0.115 | Stewardship priming narrative group[1]? Yes (1)–No (0) | -0.14 | -0.11 | ** | 0.27[D] | + | -0.24 | -0.04 |
| df = 4, 652 | ROS Personal | -0.22 | -0.22 | ** | 0.05[C] | + | -0.29 | -0.15 |
| | ROS Social | 0.12 | 0.23 | ** | 0.05[C] | + | 0.08 | 0.16 |
| Model 2: Main variables with demographic and other determinants | | | | | | | | |
| R = 0.469 | (Constant) | 2.67 | | ** | | | 2.10 | 3.25 |
| R2 = 0.22 | ROS Personal | -0.22 | -0.23 | ** | 0.05[C] | + | -0.31 | -0.13 |
| df = 39, 402 | ROS Social | 0.09 | 0.18 | ** | 0.03[C] | + | 0.04 | 0.14 |
| | How often do you consume meat in a week[2]? I don't consume meat: Yes (1)–No (0) | 0.35 | 0.17 | ** | 0.02[C] | + | 0.15 | 0.54 |
| | What is the highest level of schooling you have completed[3]? Diploma: Yes (1)–No (0) | 0.43 | 0.15 | ** | 0.02[C] | | 0.14 | 0.71 |
| | In what sort of house do you live[4]? Apartment: Yes (1)–No (0) | 0.58 | 0.14 | ** | 0.01[C] | | 0.16 | 0.99 |
| | What is the highest level of schooling you have completed[3]? Bachelor's degree: Yes (1)–No (0) | 0.18 | 0.16 | ** | 0.01[C] | | 0.05 | 0.32 |
| | What is your gross household expenses per month[5]? Above 25 million: Yes (1)–No (0) | -0.69 | -0.09 | * | 0.01[C] | | -1.35 | -0.02 |

Unsignificant result omitted. For full results see S6 File, -Multiple regression Tables 1 to 2.

*p < .05

**p < .01

[A]regression using enter method in a stepwise manner

[B]regression using enter method

[C]effect-size calculation using eta squared (F2)

[D]effect-size calculation using Hedge's g

+small effect size F2> = 0.02 (or in some cases of categorical dummy variable, using Cohen's D/Hedges'g > = 0.2); ++medium effect size F2> = 0.15 (or in some cases of categorical dummy variable, using cohen's D/Hedges'g > = 0.5)

[1]compared to respondents who fill the survey without having to read any narrative

[2]compared to respondent who eat meat once a week

[3]compared to respondent with master/PhD

[4]compared to respondent who is still live with their parents; [5]compared to those respondents whose expenses is below IDR five millions a month.

## 3.6 Demographic and other determinants

Age, gender, education, and income are often found as significant demographic determinants in most study of religion [8,10,14] and environment [11,12,61,62]. In the second regression model (Tables 10 and 11), this study informs some demographic and other determinants that closely related to outcome variables, namely weekly meat consumption, level of education, home ownership, and monthly household expenses.

## 4 Discussion

The present study supports one conclusion of White's [6] thesis whereby religion influences concerns to the natural environment through the religious teaching narrative influences towards and the relation between extrinsic social religious orientation with general environmental apathy and acceptability for harming animals. However, with intrinsic personal religious orientation, this study clearly rejects White's [6] thesis whereby we found that higher IP is more likely relates to lower GEA and acceptability for harming animals.

After examining the results, we draw several conclusion. In regards to priming narratives group treatment, this study finds only SN group significantly different from the others. The stewardship narrative group has the lowest environmental apathy and acceptability for

harming animals. This highlights the significance of the stewardship narrative to influence participants' apathy and acceptability for harming animals. *Second*, the working hypothesis of this study expects the non-narrative control group to have GEA and AIS scores somewhere in the middle between the lowest (SN) and the highest (DN). The result is true in AIS but not in GEA. The non-narrative control group have the highest apathy when compared to SN and DN. Lastly, the consistent non-significant differences between the dominance narrative and the control group show that teacher and school staff population in east java (represented by control group) seem to adopt religious teaching that is more similar to the dominance narrative type of religious ideology.

For religious orientation, both IP and ES religious orientation relate to general environmental apathy and acceptability for harming animals. Lower scores of environmental apathy and acceptability for harming animal consistently relate to higher scores of intrinsic personal and lower scores of extrinsic social religious orientations.

### 4.1 Religious narratives and environmental concerns

On the attempt to provide evidence for White's [6] thesis, the present study finds partial support. Through its narratives, religion may positively or negatively influence its follower's attitude toward the natural environment. This study's results stress the importance of communication framing to influence religious followers' interpretation towards religious scripture[26,30]. Consequently, respondents' interpretation toward a religious script may or may not influence them to adopt a specific views and attitude toward the ecology. Tables 10 and 11 show how various religious narratives influence participants' environmental concerns which is represented by general environmental apathy and acceptability for harming animals. This study finds no significance difference between human dominance narratives (DN) with non-narratives control group (NN). However, result shows consistent differences between stewardship narratives (SN) with either DN or NN. In SN group, participants consistently score the lowest GEA and AIS.

Wardekker et al. [26] argue that religious framings of climate change resonate with the electorates of both progressive and conservative politicians and could serve as bridging devices for bipartisan climate-policy initiatives. In similar studies, Feinberg & Willer [30] establish the importance of moralization as a cause of polarization on environmental attitudes and suggest that reframing environmental discourse in different moral terms can reduce the gap between liberals and conservatives in environmental concern. This study strengthens those results whereby interpretation of religious scripture influences individual's environmental concerns. Even when taking all the main variables with all the demographic and other important determinants (the second regression model), religious stewardship narrative remains a significant influence in reducing participants' apathy towards environmental concerns. Reframing religious narratives to a more responsive and considerate ideology for environmental crises reduces apathy for the natural environment and acceptability for harming animal.

### 4.2 Religious orientation to environmental concerns

White [6] marked a milestone in which research on religious allegiance toward environmental sustainability started. Ever since, more and more evidences show that religion hinders concerns for the environment [8,9] whereby religious believers relatively show lack of urgency for environmental issues. One study shows that belief in the bible consistently and independently relates to a more acceptability for exploiting the environment for economy, and to lesser concerns for air, water, and waste [63]. In other studies, end-times theology, or the belief in an afterlife, significantly relates to a lower concern for the environment [10], while lack of belief

of the afterlife or of divine intervention leads people to focus on human responsibility and the need for action, and to bolster the perceived necessities for improving the condition of the natural environment [64]. Through Allport's intrinsic personal and extrinsic social religious orientation the present study supports and rejects White's [6] thesis. On the one hand, when people intrinsically committed and view religion as their moral code, rather than hindering, religion is more likely related to a higher concerns for the natural environmental and animal protection. On the other hand however, through extrinsic social religious orientation (ES) this study finds that religion is more likely related to lower concerns for the environment. People with high scores on ES are more likely having higher acceptability for harming animals and environmental apathy. We find that ES is very close to Tajfel's social identity theory [65] in the sense that ES relates to the social identity aspects of religion (e.g. religious participation, group affiliation, etc.) [66]. Although at first ES may originates from religious belief and intrinsic commitment, it describes people motivation for social group membership of practicing religion. By affiliating to a religious group, people may gain consolation, protection and social status which in turn enable their religious participation [40–44].

Lastly through the second regression model, the present study stresses the consistent roles of religious orientation (both IP and ES) with general environment apathy and acceptability for harming animals. Even when taking all the main variables with all demographic and other important determinants as well as independently, IP and ES remain consistent in predicting the natural environment apathy and acceptability for harming animals.

## 4.3 Demographics and other determinants

In the second regression model, the present study finds that weekly meat consumption, level of education, home ownership, and monthly household expenses have significant relation to GEA and AIS (*see* Tables 10 and 11). However, after examining each of the effect-size, we need to underline that none of these relation warrant satisfying explanation.

For meat consumption, results show that participants who never consume meat in their daily diet have higher environmental apathy and acceptability for harming animals compared to those participants who only consume meat once a week. We propose to explain this result through the respondents' socio-economic status which often represented by monthly income and expenses. Respondent who answers no meat consumption in their daily intake may describe about a healthy life of their choosing; an awareness for animal rights and the environment; or may intangibly describe socio-economic factors of low monthly income and expenses. Regarding the latter, only on monthly expenses we find that participants whose expenses above IDR 25 million in a month are more likely to have lower acceptability for harming animals compared to participants whose expenses below IDR 5 million (b = -0.69, $p<0.05$). With this finding, it is tempting to conclude that the richer the participants the more likely they will have a lower acceptability for harming animals. However, we find no relation of AIS to other classes of monthly expenses category. One limitation in interpreting the data may derive from how the questionnaire provides 'refuse to answer' option in the monthly income and expenses question. Answering 'refuse to answer' to question about monthly income and expenses blurs all group difference that may be found otherwise. Therefore, this study is very limited in providing explanation about monthly income and expenses, and therefore carefully proposes further investigation to examine the relationship between monthly income and expenses with meat consumption.

For the level of education, this study finds that respondents with a Master/PhD degree have lesser acceptability for harming animals than respondents with a diploma degree. Higher education may provide more opportunities to acquire information about animal welfare and

environment preservation. Diploma education in Indonesia usually revolves around pragmatic and technical hard skills (e.g., mechanic, lab instrumentation, pharmacist, etc.) which is different from Indonesia's government education curriculum from junior high to college degree. However, this result is not consistent with other education level category.

Lastly, for respondent's type of residence, this study finds that compared to respondents who live with their parents, those respondents who live on their own have higher acceptability for harming animals. It may be that respondents who live in their own apartment are those who live in the urban settings which consequently have less interaction with animal and the natural environment in their everyday life. However, this study finds no relation between rural-urban area residence with AIS. Also, there are only eight participants in this category which clearly not enough to warrant satisfying parametric assumption.

## 5 Conclusion

In summary, this study strongly presents religion as an important key to both promote and depress environmental concerns. Contrary to White (1967) thesis, this study demonstrates how intrinsic religious orientation relates strongly to a higher environmental concern. Inherently, one aspect that religion is deeply personal and that the commitment to a religious life and living out his/her religion, is not necessarily inhibits a person's concerns for the importance of the natural environment. However, specifically in extrinsic social religious orientation, a person's motives to belong to a group or community, in the perspective of in-group membership, affiliation, providing status and consolation in context of social identity [46,65,67,68], are more likely relate to a higher environmental apathy and acceptability for harming animals.

One contribution from the present study is the compelling evidence of how religion may support or depress environmental concerns through its religious teachings. In this sense, despite the intrinsic personal and extrinsic social components of religious belief, religion's teachings and narratives influence the tendency for a person to care for animal and their natural environment.

## Supporting information

**S1 File. Treatment narratives.**
(DOCX)

**S2 File. Questionnaires in English.**
(DOCX)

**S3 File. Questionnaires Bahasa Indonesia adaptation.**
(DOCX)

**S4 File. Missing case analysis, factor analysis and reliability.**
(DOCX)

**S5 File. Manova results.**
(DOCX)

**S6 File. Multiple regression results.**
(DOCX)

**S7 File. RAWDATA_N1007.**
(XLSX)

**S8 File. SPSSDATA_N929.**
(SAV)

## Acknowledgments

We acknowledge the significance of Indonesia Endowment Fund (Lembaga Pengelola Dana Pendidikan Indonesia), the Faculty of Psychology Universitas Indonesia, Rakata Adventure and Rakata Alam Terbuka foundation, Universitas Islam Malang, Faculty of Psychology Universitas Brawijaya, Setyo Ramadi, Yeka Kusumajaya, and KH. Ahmad Zubaidah (Gus Ida), for their enduring support in various aspects of this research. We thank all the respondents for their participation in this survey.

## Author Contributions

**Conceptualization:** Dexon Pasaribu, Bagus Takwin, Pim Martens.

**Data curation:** Dexon Pasaribu.

**Formal analysis:** Dexon Pasaribu, Bagus Takwin.

**Funding acquisition:** Dexon Pasaribu.

**Investigation:** Dexon Pasaribu.

**Methodology:** Dexon Pasaribu, Bagus Takwin, Pim Martens.

**Project administration:** Dexon Pasaribu.

**Resources:** Dexon Pasaribu.

**Supervision:** Bagus Takwin, Pim Martens.

**Validation:** Dexon Pasaribu, Bagus Takwin, Pim Martens.

**Visualization:** Dexon Pasaribu.

**Writing – original draft:** Dexon Pasaribu.

**Writing – review & editing:** Dexon Pasaribu, Bagus Takwin, Pim Martens.

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
