## [Decision Letter · Decision Letter 0]

18 May 2022

PONE-D-21-24760

Can religion promote concerns for the natural environment and animal protection? The role of religious’ narrative and religious orientation toward concerns for the natural environment and animal welfare

PLOS ONE

Dear Dr. Pasaribu,

Thank you for submitting your manuscript to PLOS ONE. After careful consideration, we feel that it has merit but does not fully meet PLOS ONE’s publication criteria as it currently stands. Therefore, we invite you to submit a revised version of the manuscript that addresses the points raised during the review process.

We look forward to receiving your revised manuscript.

Kind regards,

Randeep Singh

Academic Editor

PLOS ONE

Journal Requirements:

3. Thank you for stating the following in the Acknowledgments Section of your manuscript: "We acknowledge the significance of Indonesia Endowment Fund (Lembaga Pengelola Dana Pendidikan Indonesia), the Faculty of Psychology Universitas Indonesia, Rakata Adventure, Rakata Alam Terbuka, Universitas Islam Malang, Faculty of Psychology Universitas Brawijaya, Setyo Ramadi, Yeka Kusumajaya, and KH. Ahmad Zubaidah (Gus Ida), for their enduring support in various aspects of this research. We thank all the respondents for their participation in this survey. Lastly, the work of Dexon Pasaribu is funded by Indonesia Ministry of Finance Scholarship (LPDP) (Grant no 201702221010339)."

Please remove any funding-related text from the manuscript and let us know how you would like to update your Funding Statement. Currently, your Funding Statement reads as follows: "The work by DP has been made possible by Government of Indonesia Ministry of Finance’s Endowment Fund for Education Institution’s (LPDP) (Grant no 201702221010339) scholarship and funding (https://www.lpdp.kemenkeu.go.id). 

The funder had no role in study design, data collection and analysis, decision to publish, or preparation of the manuscript."

4. Please upload a new copy of Figure 1 as the detail is not clear. Please follow the link for more information: https://blogs.plos.org/plos/2019/06/looking-good-tips-for-creating-your-plos-figures-graphics/" https://blogs.plos.org/plos/2019/06/looking-good-tips-for-creating-your-plos-figures-graphics/

5. Please include a copy of Table 21-26 which you refer to in your text on page 14.

7. Thank you for submitting the above manuscript to PLOS ONE. During our internal evaluation of the manuscript, we found significant text overlap between your submission and the following previously published works, some of which you are an author.

https://journals.plos.org/plosone/article?id=10.1371/journal.pone.0254880

Please revise the manuscript to rephrase the duplicated text, cite your sources, and provide details as to how the current manuscript advances on previous work. Please note that further consideration is dependent on the submission of a manuscript that addresses these concerns about the overlap in text with published work.

Reviewers' comments:

Reviewer's Responses to Questions

**Comments to the Author**

1. Is the manuscript technically sound, and do the data support the conclusions?

Reviewer #1: Yes

2. Has the statistical analysis been performed appropriately and rigorously? 

Reviewer #1: Yes

3. Have the authors made all data underlying the findings in their manuscript fully available?

Reviewer #1: Yes

4. Is the manuscript presented in an intelligible fashion and written in standard English?

Reviewer #1: No

5. Review Comments to the Author

Reviewer #1: There are some issues with grammar and expression in English throughout the paper that need attention

Some ideas need to be expressed more clearly. However, this is a very interesting piece of research and I encourage you to make the small changes necessary for publication

Line 295. Numbers do not add up. If 276 participants are excluded from a total of 929, then N=653, not 657. See also 464-484

353-379 I could not understand the description of the scales here. Is this a 22 item scale of 12 + 10 items, and is it separate from the 9-item GAE scale?

357 has 10 items in the anthropocentric motive measure, but there are only 9 questions in 383. Is this right?

392 Where does the AIS come from?

448-462 The conclusions from the mean scores are not justifiable. For example, "The mean score for ES was 2.82 (SD=1.01) indicating that, on the whole, the respondents have above average disposition towards viewing their religious practices as an instrument for social affiliation." This is not a valid conclusion, unless you have evidence that the mean score for a randomly selected sample of the population scores on average lower than 2.82.

6. PLOS authors have the option to publish the peer review history of their article (what does this mean?). If published, this will include your full peer review and any attached files.

Reviewer #1: **Yes: **Peter Kevern

---

## [Author Response · Author response to Decision Letter 0]

22 Jun 2022

Dear PLOS One Editor,

We would like to address two of your question:

5.Please include a copy of Table 21-26 which you refer to in your text on page 14 

We did include Table 21 to 26 in our supporting information files: S4 Missing case analysis, Factor Analysis and Reliability

6. Please review your reference list to ensure that it is complete and correct. If you have cited papers that have been retracted..

We have checked each of our reference and found no problems of retracted or anything else. However we made some changes in our references section:

a. Previously we use unpublished dissertation of Meng Jenia. We change it to googlebooks "Meng J. Origins of Attitudes towards Animals. Jenia Meng, https://books.google.com/books."

b. We shortened our reference to total of 68 references (previously 76). We removed 8 references considering that this article is the continuation of our first article and these removed reference are already covered in our first article published here in PLOS One.

Thank you.

---

## [Editor Report · Decision Letter 1]

4 Jul 2022

Can religion influence its followers’ concerns for the natural environment and animal protection? The role of religious’ narrative and religious orientation toward concerns for the natural environment and animal welfare

PONE-D-21-24760R1

Dear Dr. Pasaribu,

We’re pleased to inform you that your manuscript has been judged scientifically suitable for publication and will be formally accepted for publication once it meets all outstanding technical requirements.

Kind regards,

Randeep Singh

Academic Editor

PLOS ONE
---

## [Editor Report · Acceptance letter]

3 Aug 2022

PONE-D-21-24760R1 

The role of religious narratives and religious orientation towards concerns for the natural environment and animal welfare 

Dear Dr. Pasaribu:

I'm pleased to inform you that your manuscript has been deemed suitable for publication in PLOS ONE. Congratulations! Your manuscript is now with our production department. 

Kind regards, 

on behalf of

Dr. Randeep Singh 

Academic Editor

PLOS ONE